# Rheological engineering of perovskite suspension toward high-resolution X-ray flat-panel detector

Zihao Song[1], Xinyuan Du[1], Xin He[1], Hanqi Wang[1], Zhiqiang Liu[1], Haodi Wu[1], Hongde Luo[2], Libo Jin[2], Ling Xu [1], Zhiping Zheng[1], Guangda Niu [1,3] ✉ & Jiang Tang [1,3]

Solution-processed polycrystalline perovskite film is promising for the next generation X-ray imaging. However, the spatial resolution of current perovskite X-ray panel detectors is far lower than the theoretical limit. Herein we find that the pixel level non-uniformity, also known as fixed pattern noise, is the chief culprit affecting the signal-to-noise ratio and reducing the resolution of perovskite detectors. We report a synergistic strategy of rheological engineering the perovskite suspensions to achieve X-ray flat panel detectors with pixel-level high uniformity and near-to-limit spatial resolution. Our approach includes the addition of methylammonium iodide and polyacrylonitrile to the perovskite suspension, to synergistically enhance the flowability and particle stability of the oversaturated solution. The obtained suspension perfectly suits for the blade-coating process, avoiding the uneven distribution of solutes and particles within perovskite films. The assembled perovskite panel detector exhibits greatly improved fixed pattern noise value (1.39%), high sensitivity ($2.24 \times 10^4$ μC $Gy_{air}^{-1}$ $cm^{-2}$), low detection limit (28.57 nGy$_{air}$·s$^{-1}$) as well as good working stability, close to the performance of single crystal detectors. Moreover, the detector achieves a near-to-limit resolution of 0.51 lp/pix.

X-ray flat-panel detectors (FPDs) play critical roles in medical imaging, non-destructive inspection as well as industrial and biomedical applications[1-3]. There are two prevailing classes of X-ray FPDs, indirect detectors and direct detectors. Indirect detectors, which have dominated the market, rely on scintillators to convert X-rays into visible photons for detection but suffer from low spatial resolution[4,5]. In contrast, direct detectors are based on semiconductors to directly convert X-rays into electrical signals and are advantageous in high spatial resolution X-ray imaging[6]. However, the material choice for direct detection is limited, and the current direct detection semiconductors Si and a-Se are restricted by their low absorption capability of X-rays[6], and CdZnTe suffers from the rigorous synthesis requirements[7].

Recently, metal halide perovskites (MHPs), as newborn direct-conversion semiconductors, have drawn extensive attention to researchers in the X-ray detection field due to their high X-ray attenuation coefficients, low trap density, and long carrier diffusion length[8-10]. Both single-crystal and polycrystalline film devices have demonstrated excellent detection performance[11-13]. Compared with single crystals, polycrystalline films show great superiority in flat-panel detectors because of their large-area manufacturing capability. In view of this, many researchers are committed to the utilization of perovskite polycrystalline thick films in X-ray FPDs[4,6,14,15]. Nevertheless, the spatial resolution of perovskite X-ray FPDs deviates far from the theoretical value in previous studies[4,6]. The imaging capability of perovskite direct detectors has not been fully exploited. For example,

[1]Wuhan National Laboratory for Optoelectronics (WNLO) and School of Optical and Electronic Information, Huazhong University of Science and Technology (HUST), 430074 Wuhan, China. [2]iRay Technology Company Limited, 201206 Shanghai, China. [3]Ezhou Industrial Technology Research Institute of Huazhong University of Science and Technology, 436060 Ezhou, China. ✉e-mail: guangda_niu@hust.edu.cn

Yong Churl Kim and co-workers demonstrated a resolution of 0.22 lp pix$^{-1}$ at a modulation transfer function of 0.2 (3.1 lp mm$^{-1}$ @ 70 μm pixel), and Sarah Deumel and co-workers achieved a resolution of 0.16 lp pix$^{-1}$ (3.3 lp mm$^{-1}$ @ 50 μm pixel). In sharp contrast, the theoretical resolution limit considering the charge diffusion effect is around 0.5 lp pix$^{-1}$ [16,17] (see detailed calculation process in Supplementary Note 1). The underlying mechanism of the non-ideal spatial resolution for perovskite X-ray FPDs is not clear. While direct X-ray detection with perovskite has been demonstrated, it is still a grand challenge to achieve near-to-limit resolution for perovskite X-ray detectors.

We note that pixel-level uniformity may be a decisive factor affecting the imaging resolution from the lessons of conventional image sensors[18,19]. The pixel-level non-uniformity is also called "salt-and-pepper" fixed pattern noise (FPN), which has not been studied in perovskite X-ray detectors. Non-uniform noise seriously affects the imaging quality and reduces the signal-to-noise ratio (SNR) and the detector resolution, which is the bottleneck restricting all kinds of image sensors from reaching the background limit[20–25]. We find that the FPN values of the previous perovskite X-ray FPDs are as high as 12%[6] and 46%[4], much higher than conventional image sensors (<0.5% for visible light sensors[26] and <3% for infrared sensors[27]), and the pixel-to-pixel uniformity should be urgently improved.

In this work, we report rheological engineering of the perovskite precursors to achieve X-ray FPDs with pixel-level high uniformity and near-to-limit spatial resolution. Our approach includes the addition of methylammonium iodide (MAI) and polyacrylonitrile to the perovskite precursor solution, to synergistically enhance the flowability and particle suspension stability of the oversaturated solution. The obtained suspension perfectly suits for the blade-coating process, avoiding the uneven distribution of solutes and particles. We termed the obtained suspension as blading-grade suspension (BGS). As a result, the achieved FPDs demonstrate excellent pixel-level uniformity, and the FPN value is improved to 1.39%, much better than the control sample (15.81%). More importantly, the spatial resolution reaches 0.51 lp pix$^{-1}$ at modulation transfer function (MTF) = 0.2, which is the highest among all the reported perovskite FPDs and even close to the theoretical value (0.56 lp pix$^{-1}$). Our work points out that pixel-level uniformity is the key to the spatial resolution of perovskite X-ray FPDs, and promotes perovskite detectors closer to practical applications.

## Results

We adopted the blading-based printing method to prepare perovskite thick films, and this method is convenient to scale up. It should be noted that to achieve hundreds of micrometer-thick films, the precursor suspension includes excessive perovskite particles in the solution, which is in sharp contrast to the transparent solution for solar cell fabrication. As observed in previous studies[6,14] and our own experiment, the suspension exhibits large viscosity and thus the uneven distribution of suspensions along the printing direction renders poor uniformity within the film plane. Moreover, the pre-formed perovskite particles in the suspension could easily sediment onto the substrate, resulting in non-uniformity along the thickness direction. Thus, it is of great necessity to prepare perovskite suspensions with low viscosity and high particle suspension stability to realize uniform perovskite thick films. According to the theory of suspension rheology, particles in a liquid act as obstacles hindering the liquid's flow and then increasing the flow resistance, i.e., the viscosity[28]. It is obvious that for a given solid fraction, the smaller particles have a large surface area and thus result in higher viscosity than larger particles. This theory inspires us to increase the particle size in the perovskite suspension to decrease the viscosity.

In our experiment, we dispersed perovskite methylammonium lead iodide (MAPbI$_3$) in γ-butyrolactone (GBL) and the concentration reached 4.5 mol/L. In this oversaturated suspension, parts of perovskites are dissolved, and most of the solids are just dispersed in the solvent. The following chemical equilibrium exists in the suspension:

$$(1)$$

Both iodide anions and GBL solvents serve as the coordinators for Pb$^{2+}$, resulting in the formation of adducts with different ratios of GBL and MAI, such as (MA)$_8$(GBL)$_x$Pb$_{18}$I$_{44}$, (MA)$_2$(GBL)$_2$Pb$_3$I$_8$ and MAPbI$_3$. The composition and structure of the intermediate adducts have been verified in previous studies[29–31]. In our suspensions, we observed two kinds of particles according to the transparency (Fig. 1d and Fig. S1a, b), transparent adducts and black adducts. The transparent ones are attributed to (MA)$_8$(GBL)$_x$Pb$_{18}$I$_{44}$ and (MA)$_2$(GBL)$_2$Pb$_3$I$_8$ due to their large bandgaps, and these particles exhibit a small size of 1–10 μm. The black adducts are MAPbI$_3$, and the particles show a relatively large size of 20–60 μm. The size difference between the two adducts stems from the structural dimensionality, where 3D MAPbI$_3$ can grow into large particles much easier than 1D (MA)$_8$(GBL)$_x$Pb$_{18}$I$_{44}$ and 0D (MA)$_2$(GBL)$_2$Pb$_3$I$_8$. Hence, in order to increase the particle size in the perovskite suspension, excessive MAI could be introduced to push the equilibrium toward the MAPbI$_3$ side (Fig. 1a and Fig. S1). Then we optimized the amounts of MAI as 30 mg in 1 mL suspension, and the incubation of the suspension under stirring and heating at 85 °C helped the growth of large particles. After MAI addition, the suspension could move easily upon the direction change of the vessels. We used a rotational viscometer to quantify the viscosity of suspensions with a rotation speed of 30 rpm. The viscosity values have indeed decreased from over 10$^5$ mPa·s for the pristine suspension to 547 mPa·s for the suspension with excessive MAI.

However, the increase of the particle size inevitably exacerbates the sedimentation speed. We could observe clear supernatant in less than 1 min after the stirring was stopped (Fig. 1b and Fig. S2). Here we introduced molecular ligands to stabilize the particles, while this strategy has been widely utilized in colloidal chemistry[32,33]. The ligands could also serve as defect passivator for the formed perovskite particles. We compared three kinds of ligands for stabilizing the suspension, i.e. small molecule tetrabutylammonium hexafluorophosphate (TAHF), polymer with weak coordination capability polyvinylpyrrolidone (PVP) and polymer with strong coordination capability polyacrylonitrile (PAN). For TAHF and PVP, the suspension still exhibited obvious sedimentation within several seconds (Figure S3). The reason is the short length of the ligands for TAHF and ineffective attachment onto the perovskite particles for PVP. In comparison, the addition of PAN (30 mg/mL) could effectively enhance the suspension stability. The precursor could withstand up to 5 min without particle sedimentation, and this time duration is enough for the whole blading process. Moreover, the low viscosity of the suspension has been maintained (997 mPa·s) (Fig. 1b). Adding more PAN could further enhance the suspension stability but would lead to ineffective charge transport in the prepared film.

Figure 1c shows the preparation process of the suspension. After ball milling of PbI$_2$ and MAI in GBL solvent, we could obtain the pristine suspension. This suspension could be stored for as long as several weeks (Fig. S4), and we termed it as "storage-grade suspension". Before the blading process, excessive MAI and PAN were in sequence added to obtain the suspension with suitable rheological properties, obtaining

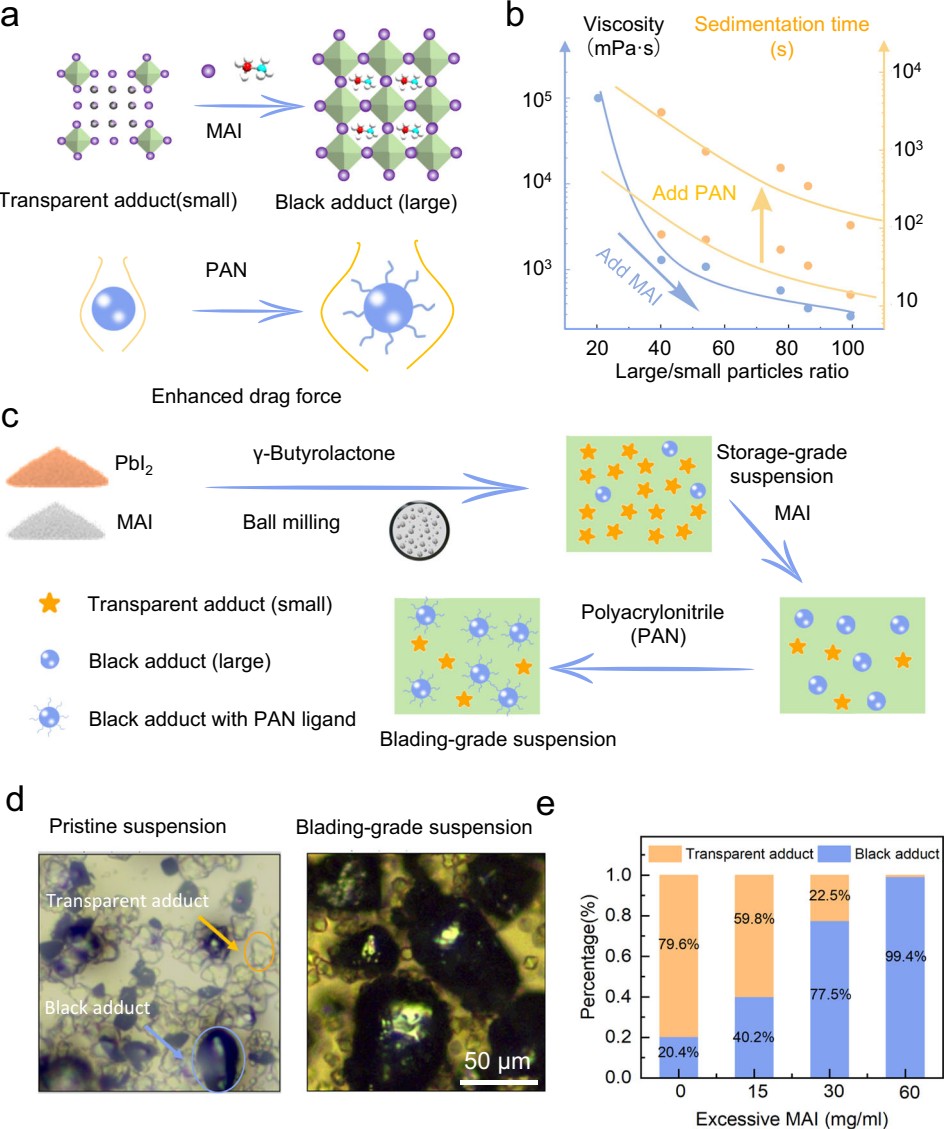

**Fig. 1 | Rheological engineering of perovskite suspension. a** Mechanism illustration of the adduct conversion and particle stabilization with Polyacrylonitrile (PAN) in methylammonium lead iodide (MAPbI$_3$) suspension. White, red, cyan, and purple balls represent H, C, N, and I atoms, and PbI$_6^{4-}$ octahedron is green. **b** The change of the viscosity and sedimentation time by the change of the particle composition. The sedimentation time was defined as the time required to form 1-mm thick supernatant after stopping the stirring. **c** The synthesis process of the blading-grade suspension. **d** Microscope images of the pristine and blading-grade suspension (excessive 30 mg/mL MAI). **e** The statistic ratio of the transparent and black adducts in the suspension.

the so-called "blading-grade suspension". The storage-grade and blading-grade suspensions guarantee the scalability of our protocol.

Then we bladed the suspension onto the indium tin oxide (ITO) and thin film transistor (TFT) substrates for electrical property evaluation. The blading speed was optimized at 0.8 cm/s, and the whole duration blading a large area of 5 × 5 cm$^2$ took 6.25 s, which is much shorter than the suspension stability duration (5 min). The cross-sectional scanning electron microscope (SEM) images of the films prepared from the pristine suspension and blading-grade suspension is shown in Fig. 2a. Obviously, the film from the pristine suspension exhibits uneven distribution along the plane direction (thickness difference) and thickness direction (particle size difference). In contrast, the film from blading-grade suspension demonstrates excellent uniformity at both directions. The schematic illustration of the blading process is shown in Fig. 2b. The sedimentation of particles results in the uneven distribution of particles and solvent during the blading process. The particles are rich at the beginning side of the blading process, while the solvents are rich at the ending side. Due to the

solubility difference between MAI and PbI$_2$, the uneven distribution of solvent also led to the uneven distribution of MAI and PbI$_2$ and thus phase separation of MAPbI$_3$ and PbI$_2$ (Fig. S5). The use of blading-grade suspension could guarantee not only the film uniformity but also the phase purity of perovskites.

We characterized the pixel-level uniformity of the perovskite films. We used special pixelated substrates with given size of pixels (1 × 1 mm$^2$, 500 × 500 μm$^2$, 200 × 200 μm$^2$, 150 × 150 μm$^2$, and 100 × 100 μm$^2$). Both single-pixel structure and pixel arrays were prepared. Figure 2c, d shows the normalized dark and X-ray response signals of the perovskite films. The measurement data are provided in Fig. S6. Both the dark and X-ray response signals of the film from pristine suspension significantly decreased upon the increase of the pixel size. The statistical results are shown in Fig. S7 to avoid the influence of coincidence. We anticipate that the fast sedimentation of large particles in pristine suspension results in the formation of voids at the interface between perovskite film and the pixel electrodes (Fig. 2e). In large pixels, the voids lead to the ineffective electrical

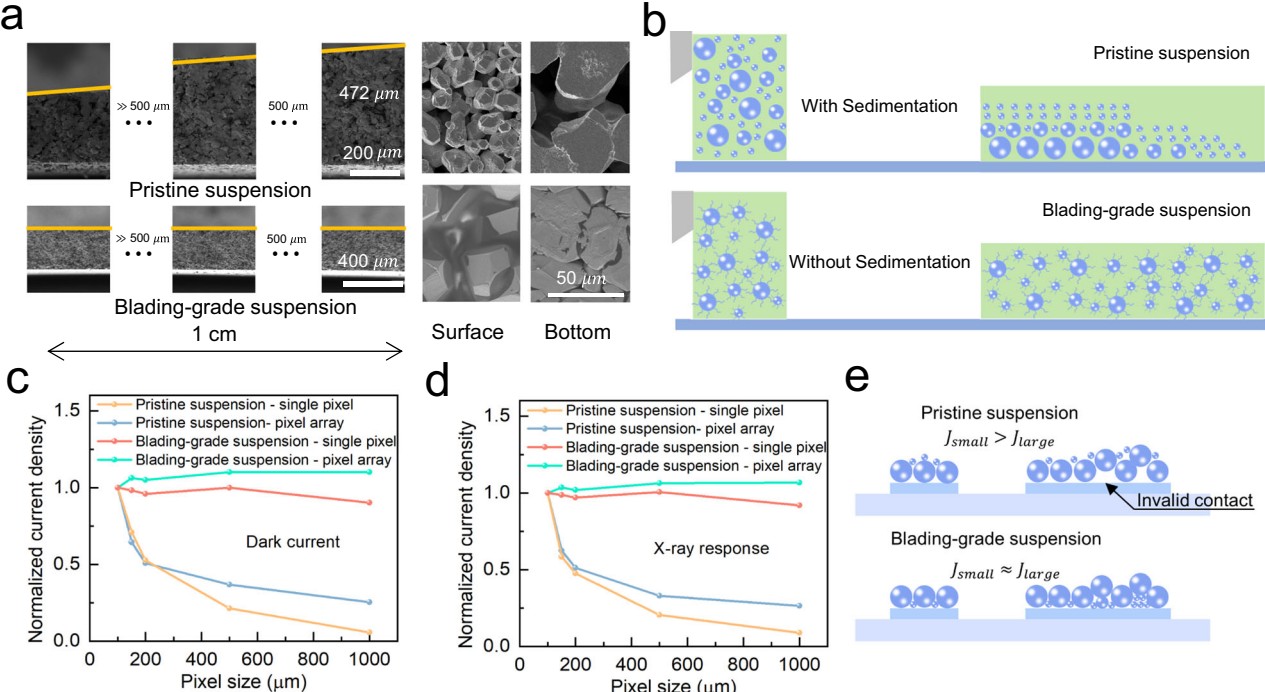

**Fig. 2 | The influence of the suspension state on the film uniformity. a** Cross-sectional, surface, and bottom SEM images of the films prepared with pristine suspension and blading-grade suspension. **b** Schematic illustration of the particle distribution during the blading process. **c** The normalized dark current densities for different pixel sizes. **d** The normalized X-ray response current densities for different pixel sizes. The devices were exposed to X-ray sources at 50 kV with a dose rate of $28.32\,\mu Gy_{air}\cdot s^{-1}$. **e** Schematic illustration of the influence of voids on the electrical contact.

connection and thus low signal amplitude. In small pixels, the voids would cause either the electrical disconnection or the negligible influence. We indeed observed more failed pixels with no signal output or extremely low output from pristine suspension for small pixels (Fig. S8). The blading-grade suspension greatly improved the pixel-level uniformity (Fig. 2c, d) and also the yield of the working devices, especially at small pixel sizes.

The basic properties and X-ray detection performance of the fabricated MAPbI$_3$ thick films are shown in Fig. 3. For the time-resolved photoluminescence spectra (Fig. 3a), there are two decay components, with the short component from the grain boundary and the long component from the grain interiors[34]. Both the short and long components of the film from blading-grade suspension ($\tau_1 = 79.06$ ns, $\tau_2 = 958.86$ ns) are much longer than that from pristine suspension ($\tau_1 = 3.96$ ns, $\tau_2 = 105.66$ ns). The $\mu\tau$ products have also been measured, where $\mu$ is the carrier mobility and $\tau$ is the lifetime. We note that the absolute $\mu\tau$ value highly depends on the light intensity due to the presence of photoconductive gain effect, and herein we used the same light intensity to compare the influence of suspension on the $\mu\tau$ results of the film. The photoconductivity curves were fitted by the modified Hecht equation (Supplementary Note 2). As shown in Fig. 3b, the film from pristine suspension held a $\mu\tau$ product of $6.79 \times 10^{-4}$ cm$^2$ V$^{-1}$. For that from blading-grade suspension, the $\mu\tau$ product boosted to $5.31 \times 10^{-3}$ cm$^2$ V$^{-1}$. The greatly increased $\mu\tau$ product can upgrade the extraction efficacy of the radiation-induced carriers within the thick film. All the above confirmed the suppressed defects for the film from blading-grade suspension, which was from the passivation effect of PAN at the grain boundaries and phase purity within the bulks. A comparison with previously reported perovskite materials shows that the measured $\mu\tau$ value is one order higher than printed MAPbI$_3$ ($1.0 \times 10^{-4}$ cm$^2$ V$^{-1}$)[6] and soft-sintered MAPbI$_3$ flat-panel detectors ($4 \times 10^{-4}$ cm$^2$ V$^{-1}$)[4], and is even close to the optimized single crystal detector (Cs$_{0.1}$FA$_{0.85}$GA$_{0.05}$Pb(I$_{0.9}$Br$_{0.1}$)$_3$: Sr, $1.29 \times 10^{-2}$ cm$^2$ V$^{-1}$)[13]. It should be noted that the lifetime $\tau$ in $\mu\tau$ is different from the photoluminescence

lifetime. Our device exhibited a photoconductive gain effect. Photoconductive gain is caused by defects in materials that can trap electrons (or holes). These trapped charges allow opposite charges like holes (or electrons) with longer lifetimes and moving between the electrodes repeatedly[35]. Thus, the lifetime in $\mu\tau$ product is the lifetime of the opposite carriers without trapping (in most cases, majority carriers). However, photoluminescence lifetime can only reflect the lifetime of minority carriers, since the photoluminescence is caused by the recombination of minority and majority carriers and the minority carriers determine the overall rate[36]. Lin and co-workers demonstrated that with the intentional introduction of traps, the lifetime of majority carriers (10 to 100 $\mu$s) is greatly larger than the lifetime (-1 $\mu$s) of minority carriers measured in time-related photoluminescence spectra[37].

The X-ray detectors were assembled with a structure of ITO/ MAPbI$_3$ film/Au (Fig. 3c). The response current versus time curves are depicted in Fig. 3d, with the electrical field intensity at 3.3 V mm$^{-1}$ and the radiation dose rate at 113.28 $\mu$Gy$_{air}\cdot$s$^{-1}$. The device from the blading-grade suspension showed 1.4 times higher X-ray response and lower dark current density than that from the pristine suspension. The response currents under different dose rates followed a linear dependence behavior at all external biases (Fig. 3e). The detection sensitivity was derived from the linear slope of curves. Figure 3f shows the sensitivity of the BGS device and the control device at different electrical field intensities. In comparison, the sensitivity of the BGS device was higher than the control device from pristine suspension at each bias. The highest sensitivity reached $2.24 \times 10^4$ $\mu$C Gy$_{air}^{-1}$ cm$^{-2}$ for the blading-grade device at the electric field of 167 V mm$^{-1}$ (50 V), while this value was three orders larger than the commercial $\alpha$-Se detectors (20 $\mu$C Gy$_{air}^{-1}$ cm$^{-2}$)[38]. To justify the performance of the detector, we measured the sensitivity of the BGS device under the standard RQA5 X-ray spectrum. The sensitivity is generally higher than the results from previous work under the same X-ray spectrum (Fig. S9)[4]. We calculated the electron-hole pair creation energy (W) of MAPbI$_3$ as 4.53 eV, according to the empirical model by Devanathan and co-

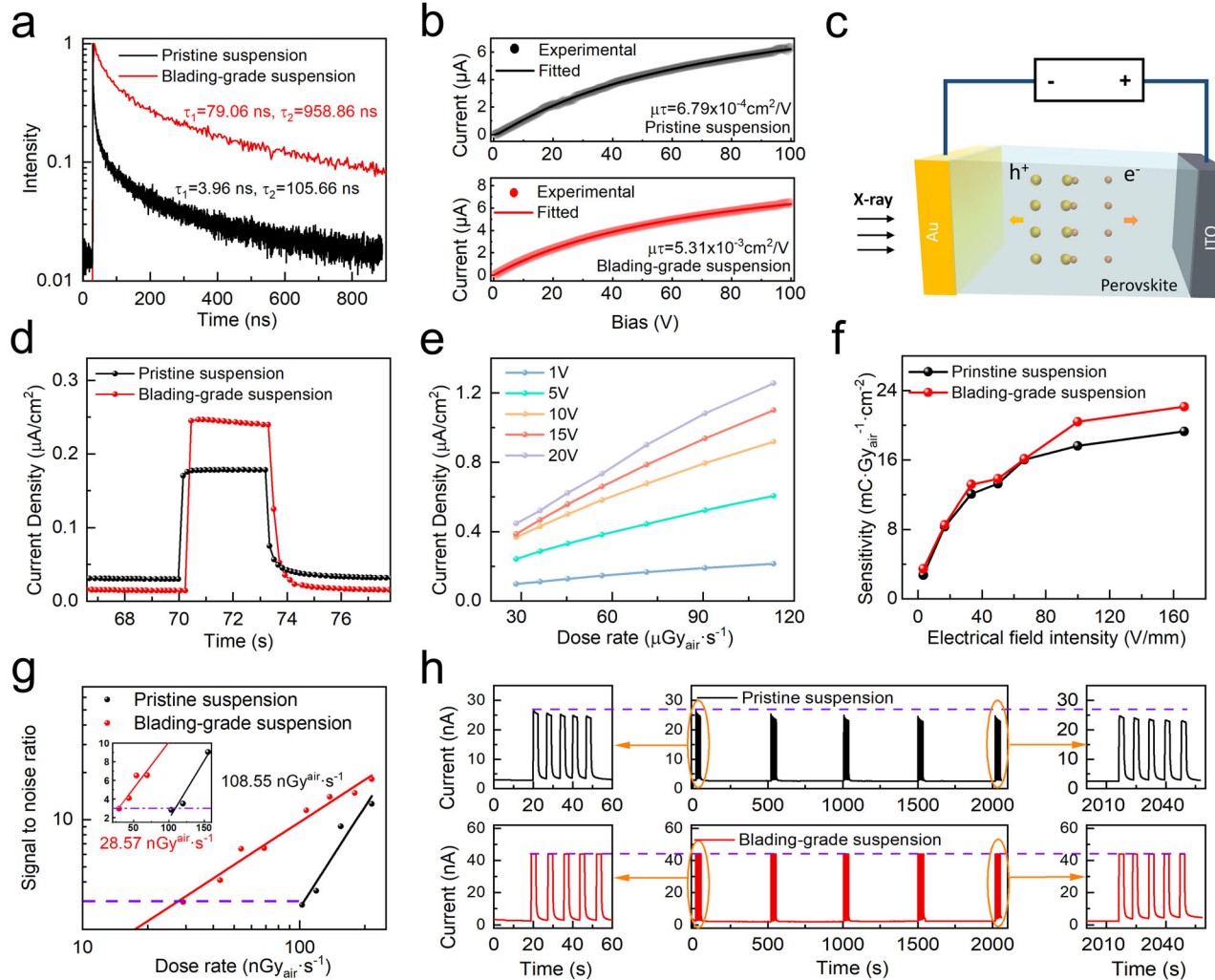

**Fig. 3 | The electrical properties of the perovskite film. a** Time-dependent photoluminescence spectra of perovskite films fabricated from different suspensions. **b** Bias-dependent photocurrent of perovskite films. The Hecht equation was applied to fit the data. **c** Device structure of the X-ray detectors. **d** The X-ray response of the devices operated at 3.3 V·mm⁻¹. The X-ray dose rate was set as 113.28 µGy$_{air}$·s⁻¹. **e** The X-ray response of the devices at different bias and X-ray dose rates. **f** The X-ray sensitivity of the devices at different bias. **g** Signal-to-noise ratio of the devices. The purple dashed line represents SNR = 3. **h** The operation stability of the devices under continuous bias and X-ray illumination.

workers[39], i.e., $W = 2E_g + 1.43$. Following our previous method[40], we derived the photoconductive gain of the detector under various biases (Fig. S12 and Supplementary Note 3).

Due to the presence of a photoconductive gain effect, the proportion of trapped carriers would increase as the dose rate decreases, resulting in nonlinear behavior, especially under low dose rates. This phenomenon could be observed in recent works[13,41,42]. To accurately obtain the detection limit, here we experimentally monitored the dose rate until SNR < 3. As shown in Fig. 3g, the BGS device achieved a low detection limit of 28.57 nGy$_{air}$·s⁻¹@SNR = 3, which was about one-fourth of the control device (108.55 nGy$_{air}$·s⁻¹) and two orders of magnitude lower than the required dosage for medical diagnostics (5.5 µGy$_{air}$·s⁻¹). The electrical stability of the detectors under X-ray radiation and bias was also recorded (Fig. 3h). The photocurrent and dark current at the electrical field intensity of 3.33 V mm⁻¹ and the radiation dose rate of 1282.4 µGy$_{air}$·s⁻¹ were monitored for 2000 s. The photocurrent of the control device decreased gradually from 23.5 nA to 19.76 nA with time, while that of the blading-grade device exhibited negligible changes from 42.08 nA to 41.61 nA.

At last, we integrated the perovskite films with the a-Si thin-film transistor arrays (Fig. 4a). The size of the TFT arrays is 50.0 × 50.0 mm with an active area of 38.4 × 38.4 mm. The pixel size is 150 × 150 µm

and the resolution is 256 × 256. Each pixel is covered with ITO and has a storage capacitance of 1.6 pF. The inactive area is protected with Si₃N₄. The common gold electrode was evaporated onto the perovskite films and served as the driving bias (Fig. 4b). The bias could be adjusted from 0 to −5V, and the electrons were extracted to the TFT arrays.

For image sensors, the pixel-to-pixel non-uniformity is called fixed pattern noise (FPN)[43]. FPN could decrease the SNR value, bringing in the deviation of the contrast and thus the spatial resolution value. According to the definition, the photo response non-uniformity (PRNU) could be represented by the standard deviation to the average response value for all the pixels within the panel detector arrays[43]. Figure 4c exhibits the dark image and X-ray image of the blading-grade film and control film. The gray-scale values of pixels were counted and fitted by the following equation (Fig. 4d).

$$y = y_0 + A \exp\left(-\frac{(x - x_c)^2}{2\sigma^2}\right) \qquad (2)$$

where $x_c$ is the average gray-scale values and σ is the standard error. Then PRNU can be calculated by the formula PRNU = $\sigma/x_c$.

The PRNU of the image greatly decreased from 15.81% for the control device to 1.39% for the BGS device. The PRNU of our BGS

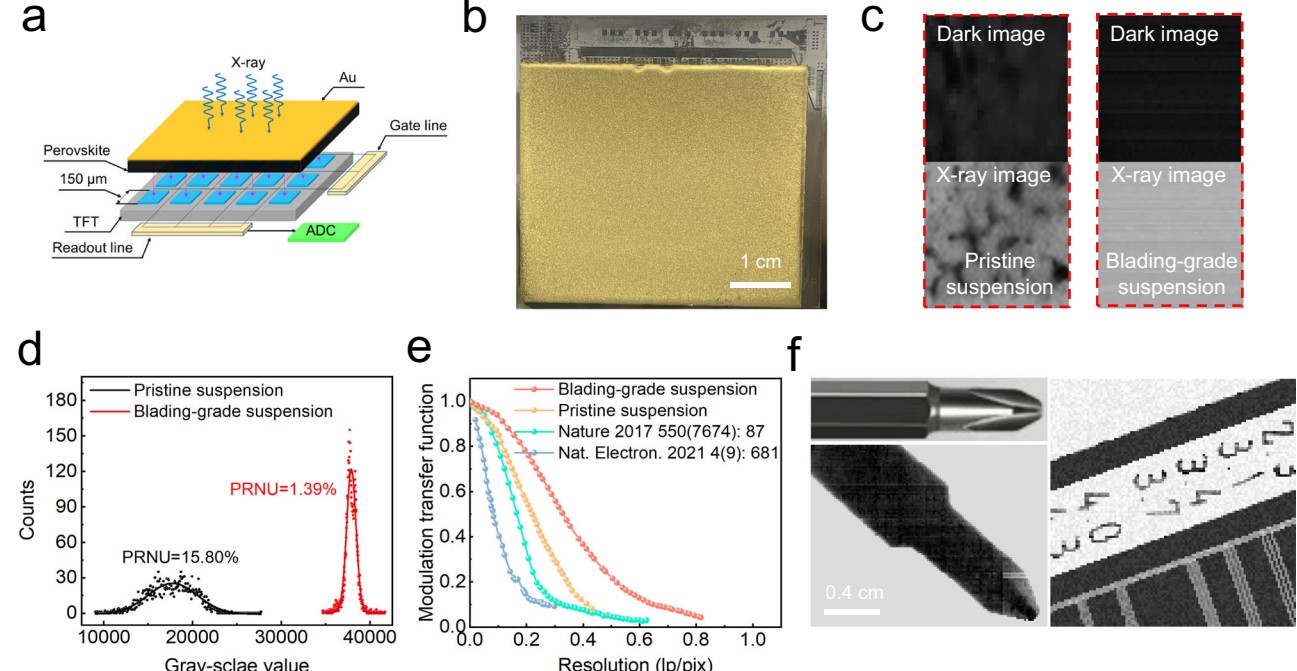

**Fig. 4 | X-ray imaging performance of the fabricated detectors. a** The structure of perovskite flat-panel detector. **b** Image of the assembled perovskite detectors onto the TFT substrate. **c** The acquired dark and X-ray images of flat-panel detectors. **d** The gray-scale value distribution of all pixels within the whole panel detectors under X-ray exposure. **e** Comparison of the spatial resolution of representative perovskite flat-panel X-ray detectors. **f** The X-ray image of the screwdriver and line pair card.

device is slightly higher than the traditional visible light detector (less than 0.5%)[26] and at the same level of HgCdTe infrared sensor (less than 3%)[27]. Compared with other X-ray detection materials, the PRNU of our detector is at the same level of α-Se (<5%) and much lower than polycrystalline $HgI_2$ film (10%) and polycrystalline CdZnTe film (20%)[21,44], demonstrating the good uniformity.

To characterize the image resolution of the blading-grade device and the control device, we measured the MTF by the slanted-edge method as shown in Fig. 4e. The unit "line pairs per pixel" (lp/pix), which is independent of pixel size, is typically utilized to accurately evaluate the imaging resolution between detectors with different pixel sizes[45]. With the improvement in uniformity, the resolution at MTF = 0.2 was increased from 0.35 lp pix$^{-1}$ to 0.51 lp pix$^{-1}$. The obtained spatial resolution is close to the theoretical limit (0.56 lp pix$^{-1}$), and further improvement has to employ an insulating grid to suppress interpixel charge diffusion within the perovskite films. In terms of resolution unit "line pairs per millimeter" (lp mm$^{-1}$), this work could achieve 3.4 lp mm$^{-1}$, which is close to that in previous works (3.3 lp mm$^{-1}$ for 50 μm pixel, 3.1 lp mm$^{-1}$ for 70 μm pixel)[4,6]. We demonstrated X-ray imaging of objects (Fig. 4f). The screwdriver head (Model PH1, diameter: 4 mm) and the standard line pair card were X-rayed with a dose rate of 3356 μGy$_{air}$·s$^{-1}$ under standard RQA3 spectrum. The lines at 3.4 lp mm$^{-1}$ could be clearly recognized, which was consistent with the resolution acquired from the MTF curve. We also derived the detective quantum efficiency (DQE) curve following the standard IEC 62220-1, and the detailed calculation process is shown in Supplementary Note 4. As shown in Fig. S13 and Table S2, the DQE is 75.3% at 0 lp mm$^{-1}$, which is close to the commercial product (80% for GC85A Samsung, 78% for 4343DXV Varex). These results show the promising potential of the uniform detector for use in non-destructive inspection and the medical field, especially for neonatal, pediatric extremities imaging and mammography where RQA3 spectrum is applied[46].

## Discussion

We found that the pixel-to-pixel signal non-uniformity of the perovskite X-ray FPDs is a critical factor for the image resolution. We developed a synergistic rheological engineering route to tune the viscosity and particle stability for the perovskite suspension, which is beneficial for uniform film fabrication. The fabricated device exhibits a high sensitivity of 2.24 × 10$^4$ μC Gy$_{air}^{-1}$ cm$^{-2}$, a low detection limit of 28.57 nGy$_{air}$·s$^{-1}$ as well as stable output. Moreover, the X-ray flat-panel detectors (FPDs) based on the uniform film achieved a near-to-limit resolution of 0.51 lp pix$^{-1}$. Further improvement of spatial resolution should focus on the charge-sharing effect between neighboring pixels. We believe this work provides a deep insight into the resolution of direct X-ray detectors and also establishes a solid foundation for the scalable and effective fabrication of high-resolution perovskite X-ray detectors.

## Methods
### Materials

All the chemicals were used as received without any further purification. Lead iodide ($PbI_2$) (99.999%) and MAI (>99.5%) were purchased from Advanced Election Technology Co., Ltd. Polyacrylonitrile (PAN) was purchased from Shanghai Macklin Biochemical Technology Co., Ltd. Tetrabutylammonium hexafluorophosphate (TADF), polyvinylpyrrolidone (PVP) and Anhydrous γ-butyrolactone (GBL) (99.9%) were purchased from Aladdin Chemical Co., Ltd.

### Suspension preparation and device fabrication

20.745 g $PbI_2$, 7.155 g MAI, and 10 mL γ-butyrolactone were put into an agate jar and balled milled at 600 rpm for 8 h. Then 2 mL viscous suspension, 60 mg MAI, and a 1 cm magneton were transferred to a 10 mL PE tube and stirred at 85 °C and 1600rpm for 8 h to get a flowable suspension. After that, 60 mg PAN was added to the unstable suspension with 30 min stirring. The suspensions are observed with an

optical microscope (LMD6) from Leica. The particles in precursors are recognized with the software ImageJ. The ratio of transparent and black adducts is calculated from the areas they cover in a picture with a size of 150 × 150 µm. The stabilized suspension was printed on a substrate and heated at 80 °C for 2 h and 100 °C for 8 h to obtain the film. The Au electrodes were prepared by a thermal evaporation method.

## Pixelated substrate

The pixelated substrate used to evaluate the electrical uniformity of film is composed of five columns and two rows of pixels. The first row contains single pixels with different sizes while the second row contains pixel arrays with different sizes. The pixels are, from left to right, one 1000 µm pixel (up) and two 1000 µm pixels (down) in 1st column, one 500 µm pixel (up) and three 500 µm pixels (down) at 2nd column, one 200 µm pixel (up) and twenty-eight 200 µm pixels (down) at 3rd column, one 150 µm pixel (up) and thirty 150 µm pixels (down) at 4th column, one 100 µm pixel (up) and thirty 100 µm pixels (down) at 5th column. The pixel area is covered with ITO while the other area is covered by $Si_3N_4$. Each pixel and pixel array are both linked to the corresponding large ITO area at the edge of the substrate by an ITO line covered with $Si_3N_4$.

## α-Si:H TFT backplane

The TFT backplane (whole area: 5 × 5 cm, active area: 3.84 × 3.84 cm) was supplied from iRay Technology Company Limited. The active area consists of 256 × 256 pixels with a pixel pitch of 150 µm. The individual pixel capacitance is 1.6 pF. The pixel area is covered with ITO while the other area is covered by $Si_3N_4$. Outside of the active area are pins of read lines and gate lines. The ROICs are supplied by iRay Technology Company Limited. The used ROICs contain a 256-channel/16-bit analog front end.

## X-ray characterization and imaging

The X-ray characterization was conducted with a medical X-ray tube (X-ray Tube Housing, Leo) purchased from Varex Inc. The X-ray spectrum was adjusted by putting 0.3 mm thick Cu films and 1 mm thick Al films in front of the X-ray tube to achieve the standard RQA3 spectrum defined by International Electrotechnical Commission guideline 62494-1. For the comparison of sensitivity with the previous study[4], we also set the tube voltage at 70 kV, and applied filtering of 0.5 mm Cu and 6.3 mm Al to achieve the standard RQA5 spectrum. The tube current can be tuned from 25 to 100 mA. A Keithley 6571B Source Meter was used to apply the bias voltage and record the response current. The X-ray dose rate was calibrated with an ion chamber dosimeter (MagicMax from IBA DOSIMETRY). The slanted-edge method specified in ISO 12233 was performed to measure the MTF curve of X-ray FPDs. A 3-mm thick rectangular lead plate was used as the edge and put longitudinal to the X-ray FPD with a 5° angle.

## Material characterization

Powder X-ray diffraction measurements were conducted on a Philips diffractometer (X'pert pro MRD) using Cu Kα radiation. The SEM tests were performed by using a Zeiss Gemini SEM 300 field-emission scanning electron microscope. The PL spectra were measured using an Edinburgh FL920 with a Xenon lamp, with the excitation wavelength at 460 nm. TRPL at 810 nm was monitored under a 478 nm light pulse as excitation from the HORIBA Scientific Delta Pro fluorimeter. For the µτ, I−V, I−t measurements, the Keithley 6571B Source Meter was used to apply the bias voltage and record the response current. A 365 nm LED light (model M365L, Zolix, Beijing) was used as the excitation source for µτ measurement. The power of 365 nm light was 420 mW. A rotational viscometer (Model NDJ−5s, purchased from LICHEN, shanghai.) was used to test the viscosity of suspensions at 30 rpm.

## Data availability

All data needed to evaluate the conclusions in this manuscript are present in the manuscript and in the Supplementary Information.

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

## Acknowledgements

This work was financially supported by the Major State Basic Research Development Program of China (2021YFB3201000), the National Natural Science Foundation of China (62134003,62074066, and 12050005), the Fund for Innovative Research Groups of the Natural Science Foundation of Hubei Province (2021CFA036, 2020CFA034), Shenzhen Basic Research Program (JCYJ20200109115212546), and the Fundamental Research Funds for the Central Universities HUST (2020JYCXJJ073, YCJJ202203001). The Innovation Foundation of Innovation Institute, Huazhong university of science and technology (5003187018). The authors also acknowledge the help from the Analytical and Testing Center of HUST.

## Author contributions

G.N. conceived and supervised the project. Z.S. designed and performed most of the experiments and analyzed the data. Z.S. and G.N. wrote the manuscript. X.D., H.W., Z.Z, L.X and J.T. helped in manuscript preparation, X.D., X.H., H.W. and Z.L. helped in the preparation and characterization of devices. X.D. helped in the simulation of charge diffusion. H. W. helped in the simulation of the modulation transfer function. H.L. and L.J. provided the TFT circuits and helped in X-ray imaging.

## Competing interests

The authors declare no competing interests.
