## [Peer Review File · Nature Communications]

REVIEWER COMMENTS

Reviewer #1 (Remarks to the Author):

The authors show a result based paper with outstanding performance of their perovskite based flat panel detectors. Most of the measurements are shown in a good and clear way, but i have some questions/remarks:

1. Figure 2 c and d: Can you provide the Measurement data behind the normalized results? The curves c and d show exactly the same behavior - i would like to know more about the measurements. Are their statistical data which can be provided or is it just one measurement of one sample? How was the data normalized and why?
2. Did you calcute the electron hole pair creation energy? Could you add it here in the paper -this would be nice.
3. Did you investigate a gain in your X-ray measurements?
4. Could you explain, why the SNR would be >0 if the dose is at 0 Gy/s
5. Could you explain why you used RQA3? For radiography, rQA5 is usually used.

Reviewer #2 (Remarks to the Author):

Rheological engineering of perovskite suspension toward high-resolution X-ray flat-panel detector

This manuscript realized the improvement of X-ray detection properties of blade-coated perovskite thick film-based X-ray detectors through rheological engineering of perovskite solution. As the blade-coating process has great potential in terms of its scalability, this strategy is very useful. On the other hand, regarding the figure-of-merit, such as the sensitivity and spatial resolution, this paper is missing proper characterization. This reviewer will suggest major revisions. Here are the comments:

(1) In the introduction part, "In view of this, many researchers are committed to the utilization of perovskite polycrystalline thick films in X-ray FPDs", the authors should cite several papers which summarized most of the perovskite-based X-ray FPDs.

Chem. Mater. 2022, 34, 12, 5323–5333

(2) Figure 1b is unacceptable. The lines/curves are from just two points or even just one point. Besides, the definition of stability is unclear. The reviewer suggests removing the light-blue background in Figure 1.

(3) The intent of Figures 2c and d is unclear.

The current density usually does not change as the pixel size changes. The authors argue that voids and/or poor contact are possible, but in this case, a single pixel may concentrate the current of multiple pixels. Furthermore, the normalization by this value not only makes comparison difficult but also gives the impression that the smaller the pixel size, the better the device. Compare using a suitable method, such as the percentage of bad pixels.

(4) Please clarify the photon source of the photocurrent for $\mu\tau$ product calculation (Figure 3b). The reviewer supposes that it is X-ray photons as shown in Figure 3c. But in this case, the $\mu\tau$ products can be overestimated as the carrier transport distance for a part of generated carriers will be shorter than the thickness.

From the lifetime (958.86 ns) and mobility-lifetime product ($5.31 \times 10^{-3} \text{ cm}^2/\text{V}$) in the manuscript, the mobility is calculated to be $5,540 \text{ cm}^2/\text{V}/\text{s}$. Even though this material is a polycrystalline film, this mobility is much higher than that of the single crystal of MAPbI₃ ($10\text{--}1000 \text{ cm}^2/\text{V}/\text{s}$).

(5) The calculation for the sensitivity is unclear. In Figure 3e, the response current increased from $0.4 \mu\text{A}/\text{cm}^2$ to $1.2 \mu\text{A}/\text{cm}^2$ at 20 V when the dose rate increased from $30 \mu\text{Gy}/\text{s}$ to $120 \mu\text{Gy}/\text{s}$, approximately. So, the sensitivity should be $0.8 \mu\text{A}/\text{cm}^2$ divided by $90 \mu\text{Gy}/\text{s}$, that is approximately $0.0089 \text{ C Gy}^{-1} \text{ cm}^{-2} = 8.9 \times 10^3 \mu\text{C Gy}^{-1} \text{ cm}^{-2}$. But the described sensitivity ($6.3 \times 10^5 \mu\text{C Gy}^{-1} \text{ cm}^{-2}$ @167V/mm) is more than 70 times higher than this value.

The manuscript needs to clarify the electric field for Figure 3g (S/N ratio) and Figure 3h (stability). As perovskite films usually suffer from ion migration problems, the signal stability should be described with the electric field.

(6)The unit of resolution that is important in this manuscript is "lp/pixel". The authors need to justify using "lp/pixel" not "lp/mm". As the FPD in this manuscript is using 150 μm pixel pitch, which is larger than that of previous reports (70 μm or 50 μm), achieving near the theoretical limit of lp/pixel should be easier compared to the previous reports. The reviewer couldn't believe that this paper made a significant improvement in terms of spatial resolution.

If the authors have achieved a higher resolution than in previous reports, please report images of a detailed object rather than the screwdriver.

Point-to-point response

Reviewer #1: *The authors show a result based paper with outstanding performance of their perovskite based flat panel detectors. Most of the measurements are shown in a good and clear way, but i have some questions/remarks:*

1. Figure 2 c and d: Can you provide the Measurement data behind the normalized results? The curves c and d show exactly the same behavior - i would like to know more about the measurements. Are their statistical data which can be provided or is it just one measurement of one sample? How was the data normalized and why?

Response: Thanks for the reviewer's recognition of our work. We provide the original measurement results and the statistical data in the new version. The measurement data for Figure 2c and d is shown in Figure R1. For Figure 2c and d, we normalized the results by dividing the current density at pixel size of 100 μm pixel, to better illustrate the variation trend. For the original data (Figure R1), the different magnitudes of the current density make the comparison difficult. Anyway, we could see that the dark current density for blading-grade suspension is much lower than the pristine suspension, which is due to the suppressed trap density by PAN in the suspension (as illustrated in Figure 3).

Figure R1. (a) The dark current densities for different pixel sizes. (b) The X-ray response current densities for different pixel sizes.

We also measured the statistical results (5 samples for each group). The films were prepared on the same pixelated substrate described in the manuscript. The results show similar trend that pristine suspension with a large variation under different pixel sizes. We have accordingly revised the

Figure R2. (a) Statistical data of the dark current and X-ray response for films from pristine suspension. (b) Statistical data of the dark current and X-ray response for films from blading-grade suspension.

2. Did you calculate the electron hole pair creation energy? Could you add it here in the paper? This would be nice.

Response: Thanks for the reviewer's suggestion. The electron-hole pair (EHP) creation energy W can be calculated according to the empirical model by Devanathan and co-authors (*NUCL INSTRUM METH A* 2006, 565, 637–649, *Nat. Photon.* 2016, 10, 333–339.):

$$W=2E_g+1.43$$

Then the electron-hole pair creation energy of MAPbI₃ ($E_g = 1.55$ eV) is 4.53 eV.

We further tried to calculate the actual EHP creation energy with the method reported by Sarah Deumel (*Nat. Electron.* 2021, 4, 681–688),

$$W_{\pm} = \frac{\text{Absorbed energy}}{\text{Numbers of extracted charges}}$$

$$\text{Absorbed energy} = \int X(E) * Aq(E)dE$$

$$Aq = 1 - \exp(-\alpha L)$$

$$\text{Number of extrated charges} = \frac{\text{Sensitivity}}{e}$$

In which, the function $Aq(E)$ was the percentage of absorbed photons (Figure R3b), which was calculated from the total attenuation coefficient α for MAPbI₃ taken from the NIST XCOM cross-

section database (Figure R3a). The function X(E) was the simulated X-ray spectrum with a RQA3 filtration (Figure R3b).

We observed that the actual EHP creation energy was lower than the theoretical value of 4.53 eV (Figure R3c). We attributed this to the photoconductive gain effect, which resulted from the photoexcited electrons trapped by defects. This enabled more holes to travel between the electrodes multiple times before recombination. As a result, many photoexcited electrons were counted repeatedly in the calculation process. We have revised the manuscript at Page13, Line 13 to 14 and Supplementary Information at Page 5.

Figure R3. (a) The attenuation coefficient of MAPbI₃. (b) The simulated X-ray spectrum with a RQA 3 filtration (black line) and the percentage of absorbed photons vs photon energy (red). (c) EHP creation energy (black) and gain of film (red) from blading-grade suspension.

3. Did you investigate a gain in your X-ray measurements?

Response: The gain can be calculated with the method documented in our previous work (*Nat. Photonics*. 2017, **11**, 726–732.),

$$G = \frac{I_R}{I_p} = \frac{J_R / (\text{Dose rate})}{J_p / (\text{Dose rate})} = \frac{S_R}{S_p}$$

$$S_p = \frac{\text{Absorbed energy}}{W_{\pm}} \times e$$

$$= \frac{6.71694 \times 10^{13} \text{ keV} \cdot \text{cm}^{-2} \cdot \text{Gy}_{\text{air}}^{-1}}{4.53 \text{ eV}} \times 1.602 \times 10^{-19} \text{ C}$$

$$= 2375.39 \mu\text{C} \cdot \text{Gy}_{\text{air}}^{-1} \cdot \text{cm}^{-2}$$

In which, I_R , I_p represent the real/theoretical photocurrent, J_R , J_p represent the real/theoretical photocurrent density, S_R , S_p represent the real/theoretical sensitivity. The theoretical sensitivity is 2375.39 $\mu\text{C} \cdot \text{Gy}_{\text{air}}^{-1} \cdot \text{cm}^{-2}$. The gain of film (red) from blading-grade suspension is shown in Figure

R2c. We have revised the manuscript at Page13 Line 15 to 16 and Supplementary Information at Page 5.

4. Could you explain, why the SNR would be >0 if the dose is at 0 Gy/s

Response: We appreciate your comment. In most of perovskite X-ray detectors, there is photoconductive gain effect (*Nat. Photon.* 2022, 16, 575–581, *Nat. Commun.* 2023, 14, 626, *J. Phys. Chem. Lett.* 2021, 12, 6961–6966.). Photoconductive gain is caused by defects in materials that can trap electrons (or holes). These trapped charges allow opposite charges like holes (or electrons) to move between the electrodes repeatedly, boosting the signal. As the light intensity decreases, the proportion of trapped charge carriers will increase and lead to nonlinear behavior especially at low dose rate.

We noticed that the SNR did not change linearly with dose rate in many published articles because of photoconductive gain. Figure R3a exhibits a representative result in a recent work (*Nat Photon.* 2022, 16, 575–581.). SNR would be >0 when the dose rate is 0 Gy s^{-1} if the data were fitted linearly. In our work, we experimentally monitored the dose rate until $\text{SNR} < 3$, and we obtained the detection limit value with $\text{SNR} = 3$. The film from blading-grade suspension exhibited the detection limit of $27.9 \text{ nGy}_{\text{air}} \text{ s}^{-1}$, and that from pristine suspension is $106.9 \text{ nGy}_{\text{air}} \text{ s}^{-1}$ (Figure R4b). We have revised the manuscript at Page 2 Line13, Page 13 Line 17-22 and Page 18 Line 3.

Figure R4. (a) The SNR versus dose rate in *Nat. Photon.* 2022, 16, 575–581. (b) The SNR versus dose rate figure of our devices with x and y axis in log style.

5. Could you explain why you used RQA3? For radiography, RQA5 is usually used.

Response: Thanks for the reviewer's suggestion. RQA3, 5, 7 were set for different purposes. Both RQA3 and RQA5 are used for general radiography (the IEC standard 62220-1). RQA3 is suitable in neonatal, pediatric extremities imaging and mammography, while RQA5 is commonly used to image extremities, head and shoulder in adults (RSNA 2003, 27710, 37-47.). We have added the description of the potential application field at Page 17 Line 1-3.

Reviewer #2: *This manuscript realized the improvement of X-ray detection properties of blade-coated perovskite thick film-based X-ray detectors through rheological engineering of perovskite solution. As the blade-coating process has great potential in terms of its scalability, this strategy is very useful. On the other hand, regarding the figure-of-merit, such as the sensitivity and spatial resolution, this paper is missing proper characterization. This reviewer will suggest major revisions. Here are the comments:*

(1) In the introduction part, "In view of this, many researchers are committed to the utilization of perovskite polycrystalline thick films in X-ray FPDs", the authors should cite several papers which summarized most of the perovskite-based X-ray FPDs. Chem. Mater. 2022, 34, 12, 5323–5333

Response: Thanks for the reviewer's appreciation. We have added the suggested reference and also updated the refs in other parts.

(2) Figure 1b is unacceptable. The lines/curves are from just two points or even just one point. Besides, the definition of stability is unclear. The reviewer suggests removing the light-blue background in Figure 1.

Response: Thanks for the good point. We have measured the viscosity of precursors with different MAI addition (15, 20, 30, 45 and 60 mg/ml) and the sedimentation time of these suspension before and after the addition of PAN. We also replace the term of stability by the sedimentation time. The destabilization time was defined as the time required to form 1-mm thick supernatant after stopping the stirring of the suspension, while longer time represents more stable suspension. We have revised Figure 1, Page 7 Line 14, Page 8 Line 10-11, Page9 line1 and Supplementary Information Page 7 in the manuscript accordingly.

Figure R5. The viscosity and destabilization time of MAPbI₃ suspension with excessive of MAI and PAN.

(3) *The intent of Figures 2c and d is unclear.*

Response: In Figure 2c and d, we normalized the results by dividing the current density at pixel size of 100 μm pixel, to better illustrate the variation trend. For the original data (Figure R1), the different magnitudes of the current density make the comparison difficult. Figure 2c and d are used to show the variation of current density on pixel size, since we notice that the poor homogeneity during film fabrication would seriously affect the pixel contact, as illustrated in Figure 2b. The blading-grade suspension could effectively reduce the inhomogeneity at small pixels. Moreover, the smaller pixels would be affected by the contact effect more seriously.

We have added the measurement data and statistical results in the revised version, as shown in above (Reviewer 1, Question 1). For the measurement data, we could see that the dark current density for blading-grade suspension is much lower than the pristine suspension, which is due to the suppressed trap density by PAN in the suspension. We also measured the statistical results (5 samples for each group). The films were prepared on the same pixelated substrate described in the manuscript. The results show similar trend that pristine suspension with a large variation under different pixel sizes. We have accordingly revised the manuscript at Page 10, Line 11 to 14 and Supplementary Information at Page 12 to 13.

(4) *Please clarify the photon source of the photocurrent for μt product calculation (Figure 3b). The reviewer supposes that it is X-ray photons as shown in Figure 3c. But in this case, the μt products*

can be overestimated as the carrier transport distance for a part of generated carriers will be shorter than the thickness.

From the lifetime (958.86 ns) and mobility-lifetime product ($5.31 \times 10^{-3} \text{ cm}^2/\text{V}$) in the manuscript, the mobility is calculated to be 5,540 $\text{cm}^2/\text{V}\cdot\text{s}$. Even though this material is a polycrystalline film, this mobility is much higher than that of the single crystal of MAPbI_3 (10–1000 $\text{cm}^2/\text{V}\cdot\text{s}$).

Response: The photon source of the photocurrent for $\mu\tau$ product calculation is UV light (365 nm, NO. M365L, Zolix). The estimated absorption depth is less than 320 nm, much lower than the film thickness (300 μm). Then we thought the derived $\mu\tau$ product was not overestimated.

To answer the second question, we need to clarify the meaning of the $\mu\tau$ product. The lifetime τ in $\mu\tau$ is different from the photoluminescence lifetime. Our photoconductive device exhibited photoconductive gain effect. Photoconductive gain is caused by defects in materials that can trap electrons (or holes). These trapped charges allow opposite charges like holes (or electrons) with longer lifetime and moving between the electrodes repeatedly. Thus, the lifetime in $\mu\tau$ product is the enhanced lifetime of the opposite carriers without trapping (in most case, majority carriers). However, photoluminescence lifetime can only reflect the lifetime of minority carriers, since the photoluminescence is caused by the recombination of minority and majority carriers and the minority determines the overall rate (ACS Appl. Mater. Interfaces 2013, 5(20), 10302-10309). In one recent work, Lin et. al. found that the lifetime of majority carriers (10 to 100 μs) is totally different from lifetime ($\sim 1 \mu\text{s}$) measured in time-related photoluminescence spectra. (Appl. Phys. Rev. 2023, 10, 011406)

Thereby, we can not simply calculate μ by dividing $\mu\tau$ with photoluminescence lifetime τ . In the following table, we also summarized the reported $\mu\tau$ product and photoluminescence lifetime values. If we divide $\mu\tau$ with photoluminescence lifetime, the calculated mobility value is also unreasonable. We have revised the manuscript at Page 12 Line 10 to 22, Page 20 Line 21 to 22 and Page 21 Line 1.

Table R1. The documented $\mu\tau$ product, photoluminescence lifetime, and derived mobility value.

Mater.	$\mu\tau$ ($\text{cm}^2 \text{V}^{-1}$)	Photoluminescence lifetime (ns)	Calculated μ ($\text{cm}^2 \text{V}^{-1} \text{s}^{-1}$)	ref
MAPbBr ₃	1.4×10^{-2}	~ 100	1.4×10^5	Nat. Photonics 2016, 10, 333–339

MAPbBr ₃	2.1×10 ⁻²	64	3.5×10 ⁵	Science Bulletin 2021, 66, 2199–2206
CsFA	8.47×10 ⁻³	208	4.1×10 ⁴	Nat. Photonics 2022, 16, 575–581 .
CsFAGA	1.09×10 ⁻²	523	2.1×10 ⁴	
CsFAGA:Sr	1.29×10 ⁻²	1059	1.2×10 ⁴	

(5) The calculation for the sensitivity is unclear. In Figure 3e, the response current increased from 0.4 $\mu\text{A}/\text{cm}^2$ to 1.2 $\mu\text{A}/\text{cm}^2$ at 20 V when the dose rate increased from 30 $\mu\text{Gy}/\text{s}$ to 120 $\mu\text{Gy}/\text{s}$, approximately. So, the sensitivity should be 0.8 $\mu\text{A}/\text{cm}^2$ divided by 90 $\mu\text{Gy}/\text{s}$, that is approximately 0.0089 C Gy⁻¹ cm⁻² = 8.9 e3 μC Gy⁻¹ cm⁻². But the described sensitivity (6.3e5 μC Gy⁻¹ cm⁻² @167V/mm) is more than 70 times higher than this value.

The manuscript needs to clarify the electric field for Figure 3g (S/N ratio) and Figure 3h (stability). As perovskite films usually suffer from ion migration problems, the signal stability should be described with the electric field.

Response: Thank you for pointing out this problem. We are sincerely sorry for that. The dose rate of Fig 3f was 28.32 $\mu\text{Gy}/\text{s}$. The bias was at 50V not 20V for the maximum sensitivity, and the thickness of perovskite film in this device was 300 μm . The calculation process is:

$$\begin{aligned} \text{Sensitivity} &= \frac{\text{photocurrent} - \text{darkcurrent}}{\text{dose rate}} = \frac{(0.8650 - 0.2295)\mu\text{A} \cdot \text{cm}^{-2}}{28.32 \mu\text{Gy}_{\text{air}} \cdot \text{s}^{-1}} \\ &= \frac{635500}{28.32} \mu\text{C} \cdot \text{Gy}_{\text{air}} \cdot \text{cm}^{-2} = 22439 \mu\text{C} \cdot \text{Gy}_{\text{air}} \cdot \text{cm}^{-2} \end{aligned}$$

The electric field for Figure 3g (S/N ratio) and Figure 3h (stability) is 3.33V/mm. The currents were measured for 2000s, the photocurrent of pristine film changed from 23.5 nA to 19.76 nA, the photocurrent of blading-grade film changed from 42.08 nA to 41.61 nA. We have revised the manuscript at Page 2 Line 13, Page13 Line 10-11, Page 14 Line 2-6 and Page 18 Line 3.

Figure R6. (a) The time dependent current density curve of the film from blading-grade suspension

at 50V (166.7 V/mm). (b) The sensitivity vs electrical field intensity curve for films from pristine suspension and blading-grade suspension.

(6) The unit of resolution that is important in this manuscript is "lp/pixel". The authors need to justify using "lp/pixel" not "lp/mm". As the FPD in this manuscript is using 150 μm pixel pitch, which is larger than that of previous reports (70 μm or 50 μm), achieving near the theoretical limit of lp/pixel should be easier compared to the previous reports. The reviewer couldn't believe that this paper made a significant improvement in terms of spatial resolution.

If the authors have achieved a higher resolution than in previous reports, please report images of a detailed object rather than the screwdriver.

Response: The resolution of detectors depends on the pixel size and other factors like the uniformity of film. The influence of pixel size follows the equation below,

$$Res(in\ lp \cdot mm^{-1}) = \frac{Res(in\ lp \cdot pix^{-1})}{pix\ size}$$

We used "lp/pixel" instead of "lp/mm", because we wanted to eliminate the influence of pixel size on the numbers that represented resolution, when we were discussing how the uniformity of film affected the resolution of detector. The theoretical maximum resolution for previous reports and our work should be 16.5 lp/mm for 50 μm pixel, 11.8 lp/mm for 70 μm pixel and 5.5 lp/mm for 150 μm pixel (our work). However, the obtained resolution was only 3.3 lp/mm for 50 μm pixel (Nature Electronics, 2021, 4, 681-688) and 3.1 lp/mm for 70 μm pixel (Nature, 2017, 550, 87-91), much lower than the theoretical value.

In our work, we could obtain the resolution of 3.4 lp/mm, although the pixel size (150 μm) is much larger than the two previous works. Additionally, we obtained X-ray imaging result of the line pair card, as shown below. The lines at 3.4 lp/mm can be clearly recognized, supporting the claim of high resolution.

Figure R7. (a) The x-ray imaging of a line pair card. (b) The MTF curve in lp/mm for blading grade suspension.

Moreover, it is practically important to study the imager with 150 μm pixel size, since many commercial products utilize pixel size around 150 μm , such as AZURE 3131Z-150 from Varex (150 μm), Pixium 2121&3030 from Trixell (153 μm). We have revised the manuscript at Page 16 Line 17-23.

Table R2. Typical commercial X-ray imagers.

Company	Product	Pixel size (μm)	Application
Vieworks	VIVIX-S 1717N	140	General Radiography
Trixell	Pixium 2121 & 3030	153	DR, C-Arm, CBCT
Varex	AZURE 3131Z-150	150	C-Arm

REVIEWER COMMENTS

Reviewer #2 (Remarks to the Author):

The manuscript has been substantially revised. I think it's great. I am Reviewer 2, and I have some comments, including a reply to Reviewer 1.

Reviewer 1's Q1 and Reviewer 2's Q3:

The data is organized and shown, and I think it is an accurate response. At least enough to demonstrate the usefulness of PAN.

The reason is stated as "which is due to the suppressed trap density by PAN in the suspension". Why does trap density in pristine suspension decrease as pixel size increases?

This is not my intention to deny the usefulness of PAN, but my intention is that if the problem or reason for films obtained by "pristine suspension" is understood, it may lead to other solutions.

Reviewer 1's Q4:

Not limited to perovskite materials, a phenomenon in which optical properties (response, lifetime, etc.) change nonlinearly with light intensity is observed. Therefore, we agree that nonlinear effects may be present in this material as well.

Therefore, it is recommended to show the fitted line in the range where linearity is maintained, rather than linearly fitting up to the Dose rate=0 point, for example, inset in Figure R4.

Reviewer 2's Q4:

We don't see the red curve in Supplementary Note 3 (Figure S3-c). The authors may be referring to Figure R3 in the response.

The light intensity dependence is described above. Therefore, it is a bit questionable to simply describe the lifetime measured at one light intensity. However, I think it is sufficient here if at least the light intensity is clearly indicated as an experimental condition and the result of benefiting from the photoconductive gain.

Also, I think that Figure 3-b should be shown by a linear vertical axis. If possible, clearly indicate the difference between fitting and experimental results.

Reviewer 2's Q6:

I generally agree with this statement, but I do not agree with determining the superiority or inferiority of the resolution based on the difference (percentage) from the theoretical limit.

The authors claim high resolution by achieving 3.4 lp/mm even under the condition that the theoretical limit is 5.5 lp/mm (150 μ m). However, both results of references and this work have almost the same resolution, around 3.3 lp/mm. It is unlikely that 150 μ m pixels would be difficult to manufacture, so the methods in the two papers being compared ((Nature Electronics, 2021, 4, 681-688) and (Nature, 2017, 550, 87-91)) are expected to yield similar values for 150 μ m pixels. So it's not a fair comparison.

Reviewer #3 (Remarks to the Author):

The authors report on a new way to prepare and stabilize a perovskite suspension for manufacturing perovskite-based x-ray detectors. By adding polyacrylonitrile (PAN) to the suspension the viscosity of the solution could be kept at a low level by simultaneously increasing the sedimentation time greatly improving the produced MAPbI₃ layer quality and its x-ray detecting properties.

The paper presents clear and well-supported results, with sufficient data and explanations to support its conclusions. Additionally, it reports significant improvements in the performance of MAPbI₃-based x-ray detectors, surpassing previously reported results. All comments from the previous reviewers have been adequately addressed and incorporated into the revised manuscript.

While the revised manuscript is overall well-written and presents clear results, there are a few issues that could be addressed to further improve the paper.

1. To accurately compare the performance of the x-ray detector reported in this paper with the results from references 4 and 6, it would be helpful to include measurements using a higher energy x-ray spectrum. Reference 4 is using a standard RQA5 spectrum while reference 6 is using an even higher energy spectrum of 100kV filtered with 3mm Al. While it is true that RQA3 is suitable in neonatal, pediatric extremities imaging and mammography, comparative measurements should be done with the same or at least a similar spectrum. Here RQA5 is the de facto standard in the industry. Furthermore, it would be helpful to add DQE measurements to add a figure of merit which can be easily compared with commercially available detectors and clearly shows the image formation capabilities of the x-ray detector.

2. While providing supplementary information to support your measurements and claims is appreciated, it is best to avoid directly referencing this information within the manuscript itself (e.g. page 7 line 8). The manuscript should be self-contained, with all the necessary information to understand its ideas and results included within the text, without the need to refer to supplementary information.

Point-to-point response

Reviewer #2 (Remarks to the Author):

The manuscript has been substantially revised. I think it's great. I am Reviewer 2, and I have some comments, including a reply to Reviewer 1.

Reviewer 1's Q1 and Reviewer 2's Q3:

The data is organized and shown, and I think it is an accurate response. At least enough to demonstrate the usefulness of PAN. The reason is stated as "which is due to the suppressed trap density by PAN in the suspension". Why does trap density in pristine suspension decrease as pixel size increases? This is not my intention to deny the usefulness of PAN, but my intention is that if the problem or reason for films obtained by "pristine suspension" is understood, it may lead to other solutions.

Response: Thanks for the reviewer's approval of our work. The trap density suppression effect of PAN has been well confirmed by the enhanced photoluminescence lifetime and $\mu\tau$ product. In terms of the relationship between trap density and pixel size, we think the reviewer may misunderstand our description (Figure 2c and d). For pristine suspension, both the dark current and X-ray response decreased as pixel size increased. This is not due to the decrease of trap density, but due to the formation of voids between perovskite film and pixel electrodes. In large pixels, the voids lead to the ineffective electrical connection and thus low signal amplitude. In small pixels, the voids would cause either the electrical disconnection or the negligible influence. We indeed observed more failed pixels with no signal output or extremely low output from pristine suspension for small pixels (Figure S11). For the pristine suspension, the large particles are susceptible to fast sedimentation and thereby more voids at the interface. By adding PAN, the large particle could be stabilized, which has been well characterized in Figure 1b.

Reviewer 1's Q4:

Not limited to perovskite materials, a phenomenon in which optical properties (response, lifetime, etc.) change nonlinearly with light intensity is observed. Therefore, we agree that nonlinear effects may be present in this material as well.

Therefore, it is recommended to show the fitted line in the range where linearity is maintained, rather than linearly fitting up to the Dose rate=0 point, for example, inset in Figure R4.

Response: Thanks for the reviewer's good suggestion. We have added the fitting line with linear response.

Figure R1. (a) The SNR versus dose rate figure of our devices.

Reviewer 2's Q4:

We don't see the red curve in Supplementary Note 3 (Figure S3-c). The authors may be referring to Figure R3 in the response.

The light intensity dependence is described above. Therefore, it is a bit questionable to simply describe the lifetime measured at one light intensity. However, I think it is sufficient here if at least the light intensity is clearly indicated as an experimental condition and the result of benefiting from the photoconductive gain.

Also, I think that Figure 3-b should be shown by a linear vertical axis. If possible, clearly indicate the difference between fitting and experimental results.

Response: Thanks for the reviewer's understanding. We have added the EHP creation energy in Figure S3c.

In terms of the $\mu\tau$ product, we have clearly added the light intensity as the experimental condition.

We added the note that the absolute $\mu\tau$ value highly depends on the light intensity, and herein we used the same light intensity to compare the influence of suspension on the $\mu\tau$ results of the film.

For Figure 3b, we have used the linear vertical axis, and provided the fitting and experimental results.

Figure R2. (a) Bias-dependent photocurrent of perovskite films. The Hecht equation was applied to fit the data.

Reviewer 2's Q6:

I generally agree with this statement, but I do not agree with determining the superiority or inferiority of the resolution based on the difference (percentage) from the theoretical limit.

The authors claim high resolution by achieving 3.4 lp/mm even under the condition that the theoretical limit is 5.5 lp/mm (150 μm). However, both results of references and this work have almost the same resolution, around 3.3 lp/mm. It is unlikely that 150 μm pixels would be difficult to manufacture, so the methods in the two papers being compared ((Nature Electronics, 2021, 4, 681-688) and (Nature, 2017, 550, 87-91)) are expected to yield similar values for 150 μm pixels. So it's not a fair comparison.

Response: We adopted the reviewer's advice to remove the discussion about the superiority or inferiority of the resolution based on the difference from the theoretical limit. The discussion in the manuscript has been revised as followings:

“With the improvement in uniformity, the resolution at MTF=0.2 was increased from 0.35 lp/pix to 0.51 lp/pix. The obtained spatial resolution is close to the theoretical limit (0.56 lp/pix), and further improvement have to employ insulating grid to suppress interpixel charge diffusion within the perovskite films. In term of resolution unit “line pairs per millimeter” (lp/mm), this work could achieve 3.4 lp/mm, which is close to that in previous works (3.3 lp/mm for 50 μm pixel, 3.1 lp/mm for 70 μm pixel)^{4,6}.”

Abstract: “Moreover, the detector achieves a near-to-limit resolution of 0.51 lp/pix.”

Conclusion: “Moreover, the X-ray flat panel detectors (FPDs) based on the uniform film achieved a near-to-limit resolution of 0.51 lp/pix.”

Reviewer #3 (Remarks to the Author):

The authors report on a new way to prepare and stabilize a perovskite suspension for manufacturing perovskite-based x-ray detectors. By adding polyacrylonitrile (PAN) to the suspension the viscosity of the solution could be kept at a low level by simultaneously increasing the sedimentation time greatly improving the produced MAPbI₃ layer quality and its x-ray detecting properties.

The paper presents clear and well-supported results, with sufficient data and explanations to support its conclusions. Additionally, it reports significant improvements in the performance of MAPbI₃-based x-ray detectors, surpassing previously reported results. All comments from the previous reviewers have been adequately addressed and incorporated into the revised manuscript. While the revised manuscript is overall well-written and presents clear results, there are a few issues that could be addressed to further improve the paper.

Response: We acknowledge the high approval of our work by the reviewer.

1. To accurately compare the performance of the x-ray detector reported in this paper with the results from references 4 and 6, it would be helpful to include measurements using a higher energy x-ray spectrum. Reference 4 is using a standard RQA5 spectrum while reference 6 is using an even higher energy spectrum of 100kV filtered with 3mm Al. While it is true that RQA3 is suitable in neonatal, pediatric extremities imaging and mammography, comparative measurements should be done with the same or at least a similar spectrum. Here RQA5 is the de facto standard in the industry. Furthermore, it would be helpful to add DQE measurements to add a figure of merit which can be easily compared with commercially available detectors and clearly shows the image formation capabilities of the x-ray detector.

Response: Thanks for the good suggestion. We have added the performance comparison under the standard RQA5 spectrum. The tube voltage was set to 70 kV. The X-ray was filtered with 0.5 mm Cu and 6.3 mm Al to achieve a HVL of 7.1 mm Al as the standard IEC 62220-1 required. As shown in Figure R3, the sensitivity of our device is generally higher than the results in reference 4 under various electric bias. Specifically, the sensitivity is $2613 \mu\text{C} \cdot \text{Gy}_{\text{air}}^{-1} \text{cm}^{-2}$ at bias of 3.3 V/mm, 31886

$\mu\text{C}\cdot\text{Gy}_{\text{air}}^{-1}\text{cm}^{-2}$ at 50 V/mm, 43704 $\mu\text{C}\cdot\text{Gy}_{\text{air}}^{-1}\text{cm}^{-2}$ at 166.7 V/mm.

Figure R3. The comparison of sensitivity of the device from blading-grade suspension with the reference results (Nature Electronics, 2021, 4, 681) under the standard RQA5 spectrum.

Moreover, we have followed the reviewer's suggestion to add DQE measurements. According to the standard IEC 62220-1, the DQE can be calculated by,

$$DQE = \frac{\phi G^2 MTF^2(\mu, \nu)}{NPS(\mu, \nu)} = \frac{d^2 MTF^2(\mu, \nu)}{\phi NPS(\mu, \nu)} = \frac{MTF^2(\mu, \nu)}{K \chi (NPS(\mu, \nu)) / d^2} = \frac{MTF^2(\mu, \nu)}{K \chi NPS(\mu, \nu)} \quad (1)$$

where Φ (photons/mm²) is the incident photon density, G is the gain of detector, MTF is the modulation transfer function, NPS is the noise power spectrum, d (Digitals) is the average gray scale of pixels, K (μGy) is the X-ray air kerma (or X-ray dose) and χ is the X-ray quanta per area per air kerma value of X-ray spectrum.

We conducted flat-field correction and averaged 100 images for calculation of NPS according to the standard IEC 62220-1. The NPS was derived from the 2D Fourier transformation of the averaged image. The normalized NPS was acquired by dividing the NPS with the power of the average gray scale as shown in **Figure R4a**. The X-ray quanta per area per air kerma value of X-ray spectrum could be given by the standard RQA spectrum (Table R1). Here we used RQA3 spectrum for imaging.

Then we could derive the DQE curve following **Equation (1)** with air kerma 525.9 μGy , and the result was shown in **Figure R4b**. As summarized in Table R2, our detector has exhibited comparable DQE with the commercial detectors, GC85A from Samsung, PaxScan 4343DXV and 4343RF from Varex.

Figure R4. (a) The normalized noise power spectrum of the image from the device prepared with blading-grade suspension. (b) The detective quantum efficiency of the device prepared with blading-grade suspension.

Table R1. X-ray Quanta per area per air kerma for different spectrum.

Spectrum	Quanta per area per air kerma (photons/ $(\text{mm}^2 \cdot \mu\text{Gy})$)
RQA3	21759
RQA5	30174
RQA7	32362
RQA9	31077

Table R2. DQE comparison.

Detector	Pixel pitch	DQE at 0lp/mm	DQE at 1 lp/mm
Our work	150 μm	75.3%	57%
GC85A, Samsung	140 μm	80 %	None
PaxScan 4343DXV, Varex	139 μm	78 %	None
4343RF, Varex	150 μm	None	62%

2. While providing supplementary information to support your measurements and claims is appreciated, it is best to avoid directly referencing this information within the manuscript itself (e.g. page 7 line 8). The manuscript should be self-contained, with all the necessary information to understand its ideas and results included within the text, without the need to refer to supplementary

information.

Response: Thank you for your advice, we avoid unnecessary direct referencing of supplementary information. To increase the consistency of the manuscript, we also rearranged the figure sequence in supporting information.

REVIEWERS' COMMENTS

Reviewer #2 (Remarks to the Author):

The revised manuscript presents clear and well-supported results, with sufficient data and explanations. Additionally, the reviewer believe that all comments from the reviewers have been adequately addressed and incorporated into the manuscript.

Therefore, the reviewer thinks it is worth publishing.

Reviewer #3 (Remarks to the Author):

The authors have significantly expanded the manuscript by incorporating additional measurements and computations, which further emphasize the notable performance enhancements of MAPbI₃-based x-ray detectors. The authors have satisfactorily responded to all of my comments, and they have also adequately addressed the feedback from the other reviewers. Again, the revised manuscript is well-written and, in my opinion, worth publishing.